# Human Mucosal IgA Immune Responses against Enterotoxigenic *Escherichia coli*

**DOI:** 10.3390/pathogens9090714

**Published:** 2020-08-29

**Authors:** Saman Riaz, Hans Steinsland, Kurt Hanevik

**Affiliations:** 1Department of Clinical Science, University of Bergen, Jonas Lies veg 87, N-5021 Bergen, Norway; saman.riaz@student.uib.no; 2Centre for International Health, Department of Global Public Health and Primary Care, University of Bergen, 5020 Bergen, Norway; 3Centre for Intervention Science in Maternal and Child Health, Centre for International Health, Department of Global Public Health and Primary Care, University of Bergen, 5020 Bergen, Norway; Hans.steinsland@uib.no; 4Department of Biomedicine, University of Bergen, 5020 Bergen, Norway; 5Norwegian National Advisory Unit on Tropical Infectious Diseases, Department of Medicine, Haukeland University Hospital, 5021 Bergen, Norway

**Keywords:** ETEC, mucosal, immunity, experimental infection, IgA, SIgA

## Abstract

Infection with enterotoxigenic *Escherichia coli* (ETEC) is a major contributor to diarrheal illness in children in low- and middle-income countries and travelers to these areas. There is an ongoing effort to develop vaccines against ETEC, and the most reliable immune correlate of protection against ETEC is considered to be the small intestinal secretory IgA response that targets ETEC-specific virulence factors. Since isolating IgA from small intestinal mucosa is technically and ethically challenging, requiring the use of invasive medical procedures, several other indirect methods are used as a proxy for gauging the small intestinal IgA responses. In this review, we summarize the literature reporting on anti-ETEC human IgA responses observed in blood, activated lymphocyte assayss, intestinal lavage/duodenal aspirates, and saliva from human volunteers being experimentally infected with ETEC. We describe the IgA response kinetics and responder ratios against classical and noncanonical ETEC antigens in the different sample types and discuss the implications that the results may have on vaccine development and testing.

## 1. Introduction

*Escherichia coli* was discovered in 1919 by the German-Austrian pediatrician Theodor Escherich and named after him [1]. Based on its tendency to cause disease and its preferred site of infection, *E. coli* can be categorized into three major groups: commensal, intestinal pathogenic, and extraintestinal pathogenic *E. coli* [2]. To date, seven different intestinal pathotypes of *E. coli* have been identified, including typical and atypical enteropathogenic (EPEC), adherent-invasive (AIEC), Shiga toxin-producing or enterohemorrhagic (STEC/EHEC), enteroinvasive (EIEC), enteroaggregative (EAEC), diffusely adherent (DAEC), and enterotoxigenic *E. coli* (ETEC) [2,3,4,5]. ETEC are defined as *E. coli* that produce one or both of the two protein enterotoxins called heat-labile toxin (LT) and heat-stable toxin (ST) [6]. ETEC may also produce one or more of several different ETEC colonization factors, which are surface proteins ETEC uses to anchor itself to intestinal cell lining [7]. ETEC was first properly described in 1971 when the pathogenic mechanisms of diarrhea mediated by the enterotoxins were identified [8,9]. ETEC infects orally, and it colonizes the small intestinal mucosa. It does not invade the intestinal epithelial cells but secretes enterotoxins that cause diarrhea by activating epithelial cell surface receptors that are normally responsible for regulating fluid content in the gut [10]. The clinical presentation of infection ranges from asymptomatic colonization to profuse watery diarrhea, sometimes combined with vomiting and abdominal cramps. Without prompt rehydration therapy, young children experiencing ETEC diarrhea may quickly die from dehydration [9]. ETEC infections and diarrhea are often widespread in low- and middle-income countries (LMICs). It mainly affects young children, but it is also a commonly encountered health problem among adult travelers and military personnel deployed to these areas [11,12]. Furthermore, ETEC sometimes causes outbreaks in high-income developed countries [13,14].

ETEC is estimated to cause around 75 million diarrheal episodes and approximately 50,000 deaths annually [15]. A vaccine against ETEC would be a rational and practical solution to reduce this large burden of morbidity and mortality. Previous studies have shown that prior ETEC infections may protect against subsequent ETEC diarrhea [16,17,18] and ETEC infection [19]. Combined with the observation that the rate of ETEC infections and diarrhea decreases with increasing age in endemic areas [20], there is hope that protective immunity can be achieved by vaccination. However, despite considerable efforts to develop ETEC vaccines these last four decades, no licensed vaccine that can offer broad, sustainable protection against ETEC has become available [17,21]. To develop protective vaccines, there is a need to identify both suitable molecular targets for the vaccines and immune responses that correlate with efficacious protection. Several promising vaccine targets have been identified, including, mainly, the ETEC colonization factors and toxins, and it is generally believed that a strong, small intestinal mucosal IgA response against these antigens is needed to protect against ETEC [21,22,23]. Being able to directly measure small intestinal IgA responses during vaccine trials would, therefore, be very valuable but this would require the use of complicated and invasive procedures. Researchers, therefore, usually measure IgA in other bodily fluids and use the results as a proxy to intestinal IgA responses. In this review, we summarize and interpret the current knowledge of human IgA immune responses to ETEC infections and suggest ways to increase our understanding of its protective effect in order to guide vaccine design.

## 2. Importance and Origins of IgA Responses

In mammals, the intestines are the largest lymphoepithelial organ, playing a significant part in the initiation and modulation of immune responses. The small intestine produces more antibodies than all other lymphoid organs combined, including the spleen and bone marrow [24,25].

Enteric pathogens infecting the gut tend to induce both systemic and mucosal immune responses, although the latter is considered to be most important in combating enteric infections [17,26,27]. In general, enteric infections tend to induce high titers of secreted mucosal IgA in the gut, with little or no concomitant increase in serum IgA [28]. Mucosal IgA is mainly produced as a result of B-cells responding to the infection after dendritic cells have activated them by presenting antigens from the infecting organism. The activated B-cells undergo clonal expansion, hypermutation, and then selection in the gut-associated lymphoid tissue (GALT), mainly in Peyer’s patches. This clonal population of B-cells (plasmablasts) then enters the bloodstream via the lymphatic system and, through the expression of homing markers on its surface, makes its way to mucosal effector sites along the gastrointestinal tract [29,30,31,32]. There, the plasmablasts differentiate into mucosal plasma cells, which start producing and secreting IgA that recognizes the antigens that first activated the B-cells [33,34]. These immunoglobulins are then actively secreted across the intestinal wall and into the intestinal mucosa (Figure 1a), where they bind to and immobilize the targeted pathogen or its virulence factors (Figure 1c).

Each mucosal plasma cell may produce one of two subclasses of IgA called IgA1 and IgA2. While monomeric IgA1 dominates in serum, mucosal plasma cells mainly produce dimeric forms of both IgA1 and IgA2, in which a peptide called J chain binds two monomers together (Figure 1d). Dimeric IgA2 is more resistant to bacterial proteases than dimeric IgA1 and is secreted at higher concentrations closer to areas of the gut where protease concentrations are high, such as in the colon [35]. In the process of being secreted into intestinal mucosa, a peptide called the secretory component (SC) usually remains attached to the dimeric IgA (Figure 1b,d), and SC further protects these secretory IgAs (SIgAs) from proteolytic degradation.

## 3. Direct and Indirect Measurements of Small Intestinal SIgA Responses

For pathogens like ETEC, the most reliable indicator of a potentially protective immune response would ideally be pathogen-specific SIgA from the small intestinal lumen. In early human experimental ETEC infection studies, Levine et al. collected jejunal fluids for use in quantitating SIgA by intestinally intubating the volunteers [36]. As an alternative to these invasive procedures, later studies have mainly relied on collecting and analyzing intestinal lavage specimens, where study participants drink a laxative until the stool is watery and clear. Although SIgA from intestinal lavage is likely to be somewhat representative of the SIgA being secreted into the small intestinal lumen, specimen collection is still relatively uncomfortable and time-consuming, and the SIgA is somewhat susceptible to proteolytic degradation during specimen collection and processing. Measuring SIgA in fecal samples has also been attempted [17], but it is unclear how the slow pass through the protease-rich colon affects the SIgA population.

For these reasons, efforts have been made to identify other easily available body fluids that have an immune response that may qualitatively and quantitatively reflect the SIgA response in the gut. SIgA has, for example, been successfully measured in saliva analyzed by ELISA [37] and by whole-cell flow cytometry [32]. Aase et al. found a strong correlation between anti-ETEC SIgA concentrations in lavage and submandibular/sublingual saliva, indicating that SIgA levels in saliva could reflect the gut SIgA immune response. Additionally, the IgA subclasses IgA1 and IgA2 were found in similar ratios in lavage samples and submandibular/sublingual saliva in each individual, indicating a similar origin of plasma cells producing IgA at both sites [32].

Another approach to assess the quantity and quality of SIgA in the gut has been to target the population of IgA-producing plasmablasts circulating in the blood. Solid-phase enzyme-linked immunosorbent spot (ELISPOT) assays were designed for quantifying the circulating plasmablasts. This assay detects the presence of antigen-specific antibody-secreting cells (ASCs), which can be used as an indirect measure of gut-directed immune responses [38,39]. These assays have been used to detect and quantify immunoreactivity to vaccine antigens in several vaccine trials [40,41], and the ELISPOT method was long considered to be the “gold standard” for evaluating antigen-specific immunity [17,42]. Subsequently, the antibody in lymphocyte supernatant (ALS) assay was developed and found to be equally sensitive to measuring the relative strength of IgA responses as ELISPOT assays [43]. In addition, it has the benefits of allowing for differentiation between old or new infections and simplified logistics, with storage of supernatants at 4 °C or −20 °C before analysis. The major disadvantage of the ALS assay is that the assay needs larger volumes of blood specimens (>10 mL) because of the need to isolate relatively large numbers of peripheral blood mononuclear cells (PBMCs), and the method is, therefore, not suitable for analysis of immune responses in infants. Nonetheless, the discussion regarding which method best reflects the gut immune response continues [17].

Quantifying pathogen-specific SIgA has traditionally been done by using colorimetric ELISA, although more sensitive methods like time-resolved fluorescence immunoassays [44] and electrochemiluminescence immunoassays [45] that require less sample material have also, more recently, been employed. Since different laboratories often develop their own protocols, use different types of assays, and usually report only SIgA results from one or two of the different sample types, a direct comparison of the results between studies is often challenging. Ideally, SIgA should be quantified by targeting the secretory component (Figure 1d), which is not found in nonsecretory IgA. However, most laboratories choose to use secondary antibodies that target the α-chain of IgA instead and assume that the detected IgA is actually SIgA. To improve the comparability of results between studies, it would help to develop universally standardized protocols for quantitating SIgA in different biological samples. In addition, we also believe it is important to standardize the presentation of antibody responses, especially for controlled human infection model (CHIM) studies. Controlling for baseline responses by estimating fold increases in antibody responses rather than reporting absolute antibody levels is helpful to reduce the effects of any differences in assay sensitivities between laboratories.

## 4. SIgA Responses to ETEC Infection

Having experienced an ETEC infection offers some protection against reinfection with ETEC, especially with the same ETEC strain [17,19,46,47]. This protection is reflected in decreased attack rates for new ETEC infections, decreased severity of symptoms, and in reduced fecal shedding of ETEC during subsequent infections. It is believed that SIgA plays an important role in this protection. The SIgA produced in response to infections with LT-producing ETEC, for example, have already been shown to be capable of inhibiting the attachment of ETEC to the intestinal wall and reducing its ability to cause diarrhea by stopping LT from binding to and activating the host’s GM1 ganglioside receptors [17,19,35].

Much of the data on IgA responses to ETEC infections has been obtained through human volunteer experimental infection studies, where known doses of well-characterized wild-type ETEC strains are given to human volunteers [48,49], with the earliest studies dating back to the 1970s [8]. Some insights have also been obtained through animal studies, especially in piglets and rabbits [50,51]. However, these animal models may have limited use for studying human ETEC infections because ETECs are usually species-specific. ETEC causing severe illness in piglets, for example, does not seem to cause disease in humans [21], thus limiting the usefulness of animal models in understanding protective human immune responses.

Human volunteer studies, which are often called controlled human infection model (CHIM) studies, have the principal virtue of providing researchers with an immunological baseline in the form of specimens taken before the volunteers are infected. Knowing the baseline anti-ETEC antibody levels for each volunteer allows the calculation of fold changes in antibody levels in response to the ETEC infections.

Performing CHIM studies allow the analyses of clinical and immunological responses to well-characterized ETEC strains with defined antigens and virulence factors in human hosts. Additionally, it reflects what happens in natural infection, where antigens present themselves in their native glycosylated form, adding another level of antigenic complexity [35]. Thus, analyses of samples from experimentally infected persons offer the full array of protective immune responses for assessment.

The majority of ETEC CHIM studies that have been undertaken during the last four decades have been conducted at the University of Texas Medical School in Houston, the University of Maryland School of Medicine in Baltimore, Johns Hopkins University in Baltimore, the U.S. Army Medical Research Institute of Infectious Diseases in Maryland, USA, and at the University of Bergen and Haukeland University Hospital in Bergen, Norway (Table 1). The 11 wild-type ETEC strains used in these studies include H10407, 214-4, B7A, E2528C-1, E23477A, LSN03-016011/A, WS0115A, DS26-1, TW10589, TW11681, and TW10722. H10407 has been used in many of these studies mainly because it readily infects volunteers even at moderate doses, and it usually causes moderate or severe diarrhea and strong immune responses.

The IgA immune responses against these strains have been studied in serum, saliva, feces, mucosal secretions, and ASC/ALS [17,26,32,47,52,53,54]. In these studies, a baseline measurement is usually made 2 to 0 weeks before experimental infection, followed by sampling at 7, 10, and 28 days afterward. Most analyses presented in these studies are serum-based, followed by analyses of ALS. Analyses of intestinal lavage or intestinal fluids have seldomly been done, probably because of the relatively complicated and uncomfortable sampling procedure, as mentioned above. Most of the CHIM studies listed in Table 1 have assessed IgA responses towards what we may call the classical ETEC antigens, including the lipopolysaccharides (LPS; O antigens), the heat-labile toxin (LT), and the ETEC colonization factors.

**Table 1 pathogens-09-00714-t001:** IgA responses reported by published human volunteer experimental ETEC infection studies.

	IgA Responses, Day of Peak (Fold Increase) (No. of Responders/No. of Volunteers)						
Antigens	Serum	ALS/ASC	Salivary	Intestinal ^a^	Strains	Dose (log10)	References
WC O78	d10(9)(5/6)				H10407	6	1978 Evans [55]
WC O78	d10(14)(6/7)				H10407	8	1978 Evans [55]
O167				ng(2)(3/5) *J	214-4	6, 8, 10	1977 Levine [56]
O148	ng(9)(20/29)				B7A	5, 6, 9	1979 Levine [19]
O25	ng(6,6)(2/6)				E2528C-1	5, 6, 9	1979 Levine [19]
O78	d10(7/7)ng				H10407	8	1982 Levine [57]
O78	d7(>2.5)(22/27)				H10407	7	2012 Darsley [58]
O78	d10(59.8)(42/44)	d7(1093)(42/44)	d10(4.5)(6/10)	d10(86.3)(5/5) *F	H10407	7.3, 8	2016 Chakraborty [17]
O78	d9(16)(9/15)	d9(125)(10/15)		d9(5.3)(9/15) *F	H10407	5	2018 Chakraborty [59]
O78	d9(10)(7/15)	d7(116)(12/15)		d9(9.3)(11/15) *F	H10407	6	2018 Chakraborty [59]
WC O6	d10(24)(29/30)				TW10589	6, 7, 8, 9	2014 Skrede [53]
WC O6		d10(5.6)(26/30)	d10(10.4)(25/27)	d10(7.7)(26/29) *L	TW10589	6, 7, 8, 9	2016 Aase [32]
LT	d30(2.5)(2/6)				H10407	6	1977 Evans [55]
LT	d10(3)(4/7)				H10407	8	1977 Evans [55]
LT	ng(28)(20/29)				B7A	5, 6, 9	1979 Levine [19]
LT	ng(8)(4/6)				E2528C-1	5, 6, 9	1979 Levine [19]
LT	d21(6/7)ng				H10407	8	1982 Levine [57]
LT (EltB)	d14(6.3)(12/15)	d10(10)(12/15) *			H10407	8, 9	2007 Coster [60]
LT (EltB)	d10(6.1)(13/15)	d7(9)(14/16) *			B7A	8, 10	2007 Coster [60]
LT	d28(3.3)(1/6)	d7(23)ng *			E24377A	8, 9	2008 McKenzie [61]
LT	d10(>4)(13/14)				LSN03, WS011	8–10	2019 Savarino [62]
LT (EltB)	ng(>2.5)(3/27)	ng(>4)(17/27)			H10407	7	2011 Darsley [58]
LT (EltB)	d10(2.5)(28/44)	d7(14.2)(35/44)	d28(1)(3/8)	d28(3.6)(4/5) *F	H10407	7.3, 8	2016 Chakraborty [17]
LT	ng(17.4)(10/11)				H10407	9	2017 Savarino [63]
LT (EltB)	ng(>2)(11/20)	d7(1.7)(6/20)			H10407	7	2018 Chakraborty [64]
LT (EltB)	d9(1.6)(3/15)	d9(2.6)(9/15)		ng(ng)(4/15) *F	H10407	5	2018 Chakraborty [59]
LT (EltB)	d28(1.63)(4/15)	d9(1.44)(5/15)		ng(ng)(4/15) *F	H10407	6	2018 Chakraborty [59]
LT	ng(>2)(5/12)				LSN03, WS011	8–10	2019 Savarino [62]
CFA/I fim	d30(2.6)(3/6)				H10407	6	1977 Evans [55]
CFA/I fim	d10(6.6)(4/6)				H10407	8	1977 Evans [55]
CFA/I	d8(6/7)ng				H10407	8	1982 Levine [57]
CFA/I	d28(40)(15/15)	d14–28(26)(14/15) *			H10407	8, 9	2007 Coster [60]
CFA/I	ng(>2.5)(10/27)	ng(>4)(18/27)			H10407	7	2012 Darsley [58]
CFA/I	d28(2)(22/44)	d7(4.1)(21/44)		d28(0.55)(0/5) *F	H10407	7.3, 8	2016 Chakraborty [17]
CFA/I	ng(8.9)(4/11)				H10407	9	2017 Savarino [63]
CFA/I	ng(ng)(1/15)	d9(2.9)(10/15)		no rise(4/15) *F	H10407	5	2018 Chakraborty [59]
CFA/I	ng(ng)(3/15)	d7(2.5)(7/15)		no rise(4/15) *F	H10407	6	2018 Chakraborty [59]
CFA/I	ng(>2)(12/20)	d7(3.3)(16/20)			H10407	7	2018 Chakraborty [64]
CFA/I(CfaB)	3m(9.6)(8/9)	d10(2.9)ng			TW11681	6,7,8	2019 Sakkestad [54]
CFAII		d7(88)(9/10) *		d7(72)(6/9) *J	E23477A	9	1994 Tacket [65]
CS1		d7(58)(4/10) *			E23477A	9	1994 Tacket [65]
CS1	d28(5.89)(7/17)	d7(66){14/17) *	ng(ng)(13/17)		E24377A	8,9	2008 McKenzie [61]
CS1 + CS3	ng(ng)(6/9)			d7(4)(2/8) *J	E24377A	8,7	1984 Levine [38]
CS3		d7(161)(9/10) *			E23477A	9	1994 Tacket [65]
CS3	d28(4.4)(7/17)	d7(53)(10/17) *	ng(ng)(10/17)		E24377A	8,9	2008 McKenzie [61]
CS5 (CsfA)	d28(5.2)(17/21)				TW10722	6,7,8,9,10	2019 Sakkestad [54]
CS6	d10(15)(5/16)	d7–10(12)(8/16) *			B7A	8,10	2007 Coster [60]
CS6 (CssA + B)	d10(1.3)(5/21)				TW10722	6,7,8,9,10	2019 Sakkestad [54]
CS17	ng(>2)(12/12)				LSNO3, WSO11	8–10	2019 Savarino [62]
YghJ		d7(7.62)18/20			H10407	7	2018 Chakraborty [64]
YghJ	d10(3.7)(17/21)				TW10722	6,7,8,9,10	2019 Sakkestad [54]
YghJ	d10{3.2)(7/9)	d7(267)ng			TW11681	6,7,8	2019 Sakkestad [54]

WC = whole cell; ng = not given; ALS = Antibody in lymphocyte supernatant; ASC = Antibody secreting cells (values from ASC assays indicated with *); ^a^ Lavage/feces/jejunal fluids; *L = lavage sample, *J = jejunal fluids; *F = feces; O = O-antigen; LT = heat-labile toxin; fim = fimbria; d = day of peak; CFA/I= colonization factor antigen I. Protein names in parentheses indicate the targeted subunits.

### 4.1. O-Antigen

Like all Gram-negative bacteria, ETEC has a lipopolysaccharide (LPS), also called endotoxin, coat on its outer membrane. An LPS molecule consists of a core oligosaccharide linked to an exposed and highly variable outer oligosaccharide, which, in *E.coli*, is called the O-antigen, often used for typing *E.coli* isolates [66,67] (Figure 2a). ETEC isolates typically produce one of approximately 20 different O-antigens [68,69,70], and only a few ETEC CHIM studies have assessed the IgA response against O-antigens, including O78 (H10407), O25 (E2528C-1), O148 (B7A), and O167 (214-4) (Table 1).

ETEC infections appear to induce an antibody response against O-antigens in a larger proportion of volunteers than any other ETEC antigens that have been tested. After experimental infection with ETEC, most volunteers appeared to have a significant increase in strain-specific anti-O-antigen IgA levels, with 3 studies reporting a proportion of volunteers with elevated IgA levels, or response ratios, of more than 90% [17,32,53]. It also induces the strongest responses, as measured in ALS, compared to any of the other ETEC antigens [17,59]. Responses against ETEC O-antigens have been studied in many sample types, and both the strength and responder ratio seem to increase with increasing doses given to the volunteers, with a few exceptions [55,59]. Chakraborty et al. reported that O-antigen IgA responses were high compared to anti-LT and -CF responses, even when low doses of H10407 were given to the volunteers [59]. It can also be seen that the serum IgA response to O-antigens peaks relatively early, at around days 7–10, which is earlier than the response to other ETEC antigens like LT and colonization factors (Table 1).

Measuring antibody responses against whole bacterial cells mostly reflects anti-O-antigen responses, although it is likely that other antigens on the surface of the infecting strain also contribute to the measured response. This could be the reason for the higher responder rates in studies that have measured IgA responses against whole cells rather than purified LPS [32,53].

### 4.2. Toxins

LT and ST are the ETEC virulence factors responsible for diarrheal disease manifestation (Figure 2b,d). Exposure to the large and immunogenic LT usually elicits a strong immune response, while ST, on the other hand, is small and nonimmunogenic. Different ETECs can produce one or both of these two toxins [71,72,73,74]. LT-only ETECs (i.e., ETECs that produce LT but not ST) seem to be more important contributors to travelers’ diarrhea than childhood diarrhea [75,76,77]. The fact that exposure to LT induces a strong immune response probably explains the apparent rapid reduction in LT-associated diarrhea following the first infections with LT-producing ETEC in young children living in ETEC-endemic areas [46]. LT is the most comprehensively studied antigen in the ETEC CHIM studies, and responses against LT have been studied in all sample types except salivary samples. Experimental infections with LT-producing strains generally result in strong anti-LT IgA immune responses, with responder ratios that are higher than for the colonization factors, but smaller than that for the O-antigens (Table 1). Between the different ETEC CHIM studies, both the fold increase in antibody levels and the day of maximum anti-LT IgA levels seem to vary widely.

In some of the CHIM studies listed in Table 1, volunteers experimentally infected with LT-producing ETEC were subsequently reinfected with ETEC of the same strain or a different strain [17,19]. Results from these studies generally show that a strong anti-LT immunity may not fully protect against diarrhea, but since the challenge strain also produced ST, which is not immunogenic [18,56], it seems likely that the apparent lack of protection is a result of the diarrheagenic activities of ST. There is currently an effort to produce vaccines based on ST [78,79], and, if successful, it is likely that a combination of ST and LT vaccine antigens could protect against the most debilitating diarrheal episodes caused by ETEC.

### 4.3. Colonization Factors

Adherence to epithelial cells plays a central part in ETEC’s ability to colonize the small intestinal cell wall. ETEC colonization factors are immunogenic protein structures that protrude from the surface of ETEC and bind to specific glycoproteins or glycolipid receptors on epithelial cells of the small intestine [80]. Although more than 20 different ETEC colonization factors have so far been described, the types of ETEC most commonly found associated with childhood diarrhea tend to produce one or more of the seven different colonization factors—colonization factor antigen I (CFA/I) and coli surface antigen 1 (CS1), CS2, CS3, CS4, CS5, and CS6 [76,81,82,83,84,85]. Many colonization factors are heteromorphic, filamentous structures called pili or fimbriae, which are usually made up of more than 1000 copies of stalk-forming structural subunits and a single tip adhesin [10,86,87,88]. Both the structural subunit and the tip adhesin bind to specific targets on the surface of enterocytes. For example, both CfaB (structural subunit) and CfaE (tip adhesin) that make up CFA/I bind to glycosphingolipids and glycoproteins on the epithelial cell surface in the small intestine (Figure 2d). Studies have also demonstrated that antibodies targeting CfaB and CfaE are able to affect fimbrial elasticity and inhibit adherence of CFA/I-producing bacteria in vitro, respectively, indicating that the colonization factors may be suitable targets for vaccine development [10,87,89]. It should also be noted that some colonization factor-specific antibodies may cross-react with other colonization factors [90,91].

In ETEC CHIM studies that have used CFA/I-producing ETEC, such as H10407, anti-CFA/I IgA responses have been shown to peak at variable time points, and, in most of the studies, the fold increase in anti-CFA/I IgA was not very high except for the study by Coster et al. [60]. They found a 40- and a 26-fold increase in serum and ALS IgA levels, respectively, and with a high response ratio for both sample types (Table 1). Immune responses to CFA/I usually peaked later than the anti-LPS response, and when volunteers who were experimentally infected with a CFA/I-producing strain were later reinfected with the same strain, their CFA/I responses were larger during the reinfection than during the initial infection [17]. The most likely explanation for this increase is that it takes more than a single infection to optimize the immune response against this antigen [47]. Furthermore, anti-CFA/I IgA responders were few or absent when fecal samples from the infected volunteers were tested [17,59]. It should also be noted that increased anti-colonization factor IgA levels in serum after infection appeared to be more frequent in volunteers who did not develop diarrhea [17,60,61,62].

In CHIM studies with ETEC strain E24377A, the volunteers developed strong IgA immune responses against the CS1 and CS3 colonization factors [38]. Responses against these factors were more elevated in ALS/ASC samples than in serum [61]. In studies involving ETEC strain TW10722, the anti-CS5 IgA responses were found to peak at 3 months after infection and remained elevated for at least 2 years [54]. Several of the strains used in CHIM studies produce CS6, which is the most common colonization factor produced by human ETEC. Although IgA responses to CS6 have been poor in most of these studies [23,60,92,93], two studies have managed to show higher levels of immunogenicity in humans after both natural infection and administration of a vaccine candidate containing CS6 [94,95].

### 4.4. Nonclassical Antigens

Beyond the classical ETEC antigens described above, newer research has identified other virulence factors that may contribute considerably to natural protection against ETEC [23,59]. In recent CHIM studies, immune responses against several of these antigens have been found to be strong, supporting the notion that these could represent useful vaccine targets. The most relevant of these include the mucinases YghJ [96] and EatA [97], the adhesins EtpA [98] and EaeH [99], the structural subunit of the flagellar shaft FliC [98], the flagellar hook protein FIgE [100], the adhesion–autotransporter proteins antigen 43 [101] and TibA [102], and the type 1 pili [57].

Many *E. coli* strains produce the YghJ metalloprotease, which can degrade major mucins MUC2 and MUC3 that make up most of the mucosal barrier that lines the intestinal tract [97]. YghJ is produced by 89% of epidemiologically relevant ETECs, and it shares extensive similarity with an accessory colonization virulence factor of *V. cholerae* [96,103]. YghJ is believed to play a pivotal role in the pathogenicity of ETEC as it helps to degrade the protective mucin layer, allowing ETECs to gain access to the epithelial cell surface [54,104].

The CHIM studies that, so far, have looked at YghJ responses have found strong IgA responses in both serum and ALS in most volunteers [54,59]. Anti-YghJ responses seem to rise and decline fast [54], probably as a result of the volunteers having been exposed to YghJ during previous *E. coli* infections.

The remaining nonclassical antigens have, so far, only been evaluated in one ETEC CHIM study [23] and are therefore not listed in Table 1. However, that study found strong ALS IgA immune responses against the flagellar antigens FliC and FlgE, as well as against antigen 43 and EatA [23]. It also reported strong responses against EtpA. EtpA is a high-molecular-weight exoprotein adhesion molecule that may bind to the tip of the ETEC flagella. The flagella, which is also called the *E. coli* H-antigen, is made up of FliC subunits, and the flagella-bound EtpA facilitates adhesion of the flagella to the intestinal epithelial cell surface, similar to the colonization factors [98] (Figure 2c). The autotransporter protein EatA is expressed by many ETEC strains and preferentially degrades MUC2. It is postulated that EatA can degrade the EtpA adhesin and, thereby, allow it to escape the host defense system by preventing it from binding too strongly to a single epithelial cell [97,104,105]. EatA has also been implicated in accelerating the delivery of LT onto the epithelial cells [106]. About 50% of all ETEC isolates secrete EtpA and/or EatA [107]. ALS IgA responses to EatA peaked at day 7 in one study analyzing a whole array of nonclassical antigens [23,64]. ALS IgA response to EaeH on day 7 was moderate in 14 out 20 volunteers, showing at least a 50% increase from baseline [59]. TibA immune responses have not been reported in an ETEC CHIM study. Finally, no antibody response against type 1 pili was detected in 7 volunteers challenged with H10407 [57].

## 5. SIgA and Protective Immunity

It is not fully understood how the intestinal SIgA immune responses to an ETEC infection actually represent a protective immunity [22,59,108]. More knowledge about which ETEC antigens contribute most to inducing a protective immune response against ETEC at the intestinal mucosal surface will help to speed up ETEC vaccine development. Although only a small number of ETEC strains have been tested in CHIM studies, and only a few of these studies have assessed intestinal SIgA responses, CHIM studies continue to shed new light on how the human immune system responds to ETEC infections [59]. To further identify which immune responses are protective, there is also a rationale for performing more CHIM studies where volunteers who have recovered from the infection are reinfected with the same or different ETEC strains. This would help to highlight the association between specific immune responses and the degree of colonization or illness. During our literature search, we came across only two published studies that included rechallenge with the same or different wild-type ETEC strains [17,19,47]. Findings from both studies support the notion that immunological protection against ETEC occurs at the mucosal surface of the small intestine. They found that reinfection with the same strain reduced the incidence of diarrhea and resulted in strong immune responses against key ETEC virulence factors. Importantly, this apparent protective immunity did not seem to stop ETEC from colonizing the small intestine of the volunteers, but fecal shedding of ETEC was 100 times lower among the volunteers who did not develop diarrhea [47]. This finding suggests that a protective intestinal SIgA response may not necessarily prevent ETEC colonization, but it may limit the severity of the colonization and toxic activities of ETEC. However, an unpublished study observed a lack of protection upon rechallenge with the B7A strain, leaving some uncertainty regarding protective immunity against a previous infection [Talaat et al., abstract number 1759, ASTMH-2017].

There seems to be an increase in IgA responses with an increasing amount of ingested ETEC or inoculation dose [19,38,56,59]. So far, however, little effort has been made to investigate this observation further. It seems beyond doubt that increasing inoculation doses caused more volunteers to develop diarrhea. It is, therefore, likely that the increased IgA responses observed when increasing inoculation doses are the result of more volunteers becoming successfully colonized rather than any direct effect of the dose. This notion is supported by findings by Sakkestad et al. [54], where experiencing diarrhea was associated with stronger immune responses even when the analyses were adjusted for the dose that the volunteers ingested. In addition, several ETEC CHIM studies have looked for, but found no clear evidence of, whether the amount of ETEC in the stools of the volunteers could be associated with dose [17,47,55,59], and the strength of the immune responses did not seem to be associated with the severity of the diarrhoeal episodes [60,109].

ETEC represents a genotypically and phenotypically diverse group of *E. coli,* and the variability in ETEC’s repertoire of virulence factors and antigenic properties is likely to contribute to the variability in immune responses that may or may not protect against new ETEC infections [60,110]. It is uncertain whether immune responses that develop as a result of natural ETEC infections will be stronger and more protective than what is possible to achieve by vaccination [61,65]. Nevertheless, there is hope that vaccinating with key ETEC virulence factor antigens will induce a strong and targeted protective immune response.

Parenteral vaccination of volunteers with type 1 somatic pili, which is commonly expressed by *E. coli*, including ETEC, was shown to cause a strong SIgA response (measured in jejunal fluids) in 12 out of 13 vaccinees, but when subsequently experimentally infected with ETEC, the vaccinees did not appear to be protected against diarrhea [21]. Using live attenuated ETEC strains as vaccines have also been attempted. The vaccine candidate based on the attenuated ETEC strain E1392-75-2A elicited strong intestinal anti-CS1 and -CS3 SIgA responses [21], but the vaccine candidate never reached clinical trials. An inactivated version of E1392-75-2A was, however, tested in a small clinical trial, but there were no signs of it being able to protect against experimental infection with a different CS1- and CS3-producing ETEC [21]. In more recent attempts, being vaccinated with the ACE527 vaccine candidate, which consists of three live attenuated ETEC strains, quantitatively reduced fecal ETEC shedding during subsequent challenge with ETEC strain H10407, and the vaccinees showed a 92% ALS response rate to the EltB subunit of LT and 56% to CFA/I when the vaccine candidate was given as a 2-dose regimen [111]. Sadly, it did not cause a significant reduction in the primary endpoint of moderate and severe diarrhea [58].

Potentially, the lack of apparent protection in these vaccine trials is due to poor antibody responses against the nonclassical ETEC antigens. For example, it has been shown that experimental H10407 infections result in a much more robust anti-YghJ antibody response than what was achieved by ACE527 vaccination [23].

In recent years, there has been an effort to develop detoxified versions of classical and nonclassical ETEC antigens that can be added to new and existing vaccine designs to bolster its protective efficacy [112]. Hence, an engineered form of LT has been produced and tested for use in such vaccines. By introducing two amino acid changes to native LT, the toxic activities of LT have been abolished without affecting its immunogenic properties [113]. This double-mutant LT (dmLT) has thus been rendered safe for use in vaccines, and it has also been shown to act as a mucosal adjuvant in addition to inducing protective anti-LT antibody responses [110,114]. Furthermore, when given parenterally, the adjuvant properties of dmLT also mediate a gut-directed immune response to the vaccine antigens, and dmLT may, therefore, become very useful for inducing a protective intestinal SIgA antibody response even for injectable ETEC vaccines [115].

A different method to offer protection against ETEC infection and disease is through passive immunization with antibodies specific to ETEC antigens [116]. This approach was encouraged by findings that oral intakes of bovine colostrum containing antibodies (mainly IgG) against the ETEC colonization factor CS17 appeared to protect volunteers by reducing the shedding of CS17-producing ETEC and providing complete protection against diarrhea [62]. Indeed, passive immunization with monoclonal SIgA antibodies is thought to be a viable approach to broad protection against ETEC infection [117].

In conclusion, the phenotypic and antigenic diversity of ETEC appears to be far more varied than initially appreciated [23,59,60], and it seems likely that including nonclassical ETEC antigens in vaccine design may improve the protective efficacy of new and old vaccine designs.

## 6. Limitations of ETEC CHIM Studies

When discussing the antibody responses reported by the ETEC CHIM studies listed in Table 1, it is also important to be aware of the limitations when comparing and interpreting results from these studies. Study sizes are usually small; ETEC strains used in the models only represent a small proportion of the ETEC population, and they have usually been selected based on their ability to colonize and cause disease in adults; the studies are conducted in an artificial and controlled setting; the study populations are always healthy adults who often differ with respect to age, race, diet, and nutritional status; different doses are used even when studying the same strain. For these reasons, it is difficult to generalize results from these studies to populations that are most at risk of experiencing ETEC infection and diarrhea, including young children in LMICs and visitors to these countries [18,58].

An intriguing finding in the overview of IgA responses in ETEC CHIM studies is that distinct antibody responses are seldomly achieved in 100% of the infected volunteers. In earlier studies, before researchers started to neutralize gastric acids before administering the dose, the actual dose that reached the small intestine was probably relatively low [47]. Although ETEC is normally detected in the stools of the volunteers throughout the designated study period, in some volunteers, ETEC shedding remains low or nondetectible throughout the study, and diarrhea does not develop [52]. Many factors probably contribute to this apparent protection against ETEC, such as hitherto undescribed host genetic factors, pre-existing protective immunity, or that these volunteers have gut microbiota providing unfavorable conditions for ETEC colonization. In fact, volunteers who are not properly colonized or who do not develop diarrhea constitute a very interesting population for studying different aspects of protection against ETEC. Since performing ETEC CHIMs is time-consuming and costly, not sampling enough to learn as much as possible about correlates of a protective immune response is a missed opportunity.

## 7. Conclusions

Experimental infection with an ETEC strain does not seem to confer any strong protective immunity against heterologous ETEC strains [19], and efforts to develop broadly protective ETEC vaccines have recently been focused on incorporating LT- and ST-based vaccine antigens, colonization factor antigens, and nonclassical ETEC antigens in the vaccine design. Currently, trials of an oral inactivated vaccine candidate are ongoing, both in travelers and LMIC children [37]. Still lacking a reliable immune correlate of protection, ALS-based assays are usually considered to be the best proxy for measuring the intestinal SIgA antibody response that is assumed to be protective. However, the recent finding that SIgA antibody levels in intestinal lavage were comparable to IgA levels in ALS and submandibular/sublingual saliva specimens [32] raises hope that saliva could represent a practical and noninvasive method to assess potentially protective intestinal SIgA antibody responses during vaccine trials. This would be especially welcome in field trials involving children.

## Figures and Tables

**Figure 1 pathogens-09-00714-f001:**
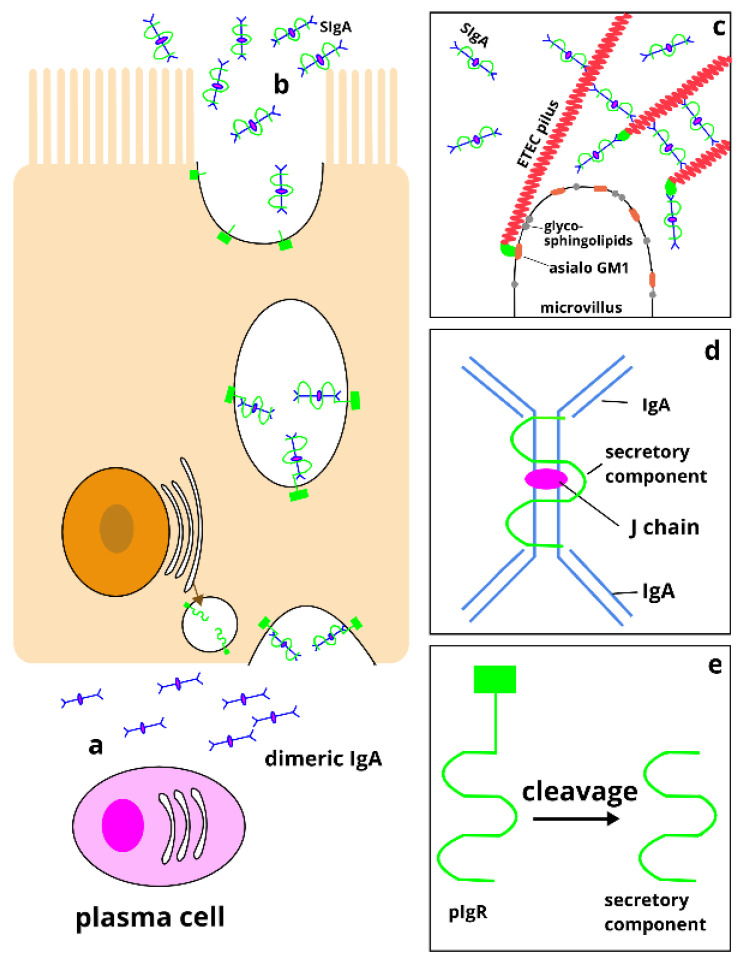
Production and secretion of secretory IgA. (**a**) IgA is secreted from mucosal plasma cells in the lamina propria (i.e., effector sites) after infection. Two IgA monomers combine with a J chain peptide to form IgA dimers before binding to the polymeric immunoglobulin receptor (pIgR) that is expressed on the basolateral surface of epithelial cells and responsible for transport across the epithelium in a process known as transcytosis. (**b**) The pIgR-bound dimeric IgA is transferred to the mucosal surface where most of the pIgR is cleaved off and dimeric IgA is released. The part of pIgR that remains is known as the secretory component (SC), and a dimeric IgA that has SC attached is known as a secretory IgA (SIgA) [35]. (**c**) Example of how SIgA may neutralize virulence factors. The figure shows the adhesion of one ETEC colonization factor fimbria anchored to the intestinal cell surface. SIgAs that recognize these fimbrial proteins block the binding of two other fimbriae. (**d**) Schematic diagram showing SIgA, a dimeric IgA antibody with J chain and secretory component. (**e**) Cleavage of pIgR at the apical surface of epithelial cells leads to the release of the secretory component that remains attached to SIgA.

**Figure 2 pathogens-09-00714-f002:**
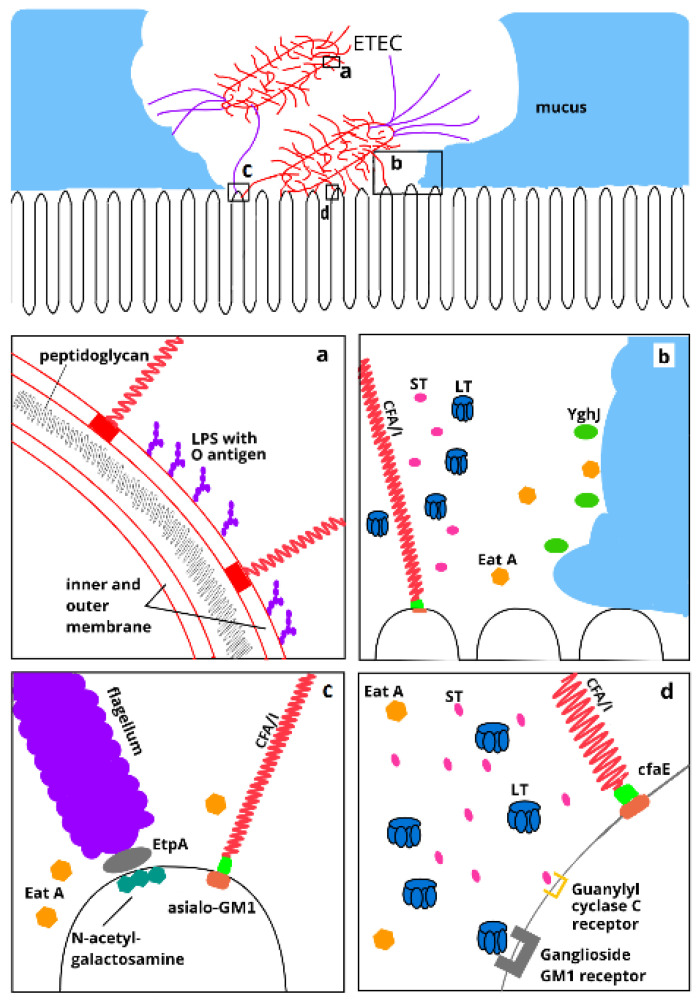
ETEC antigens. (**a**) Illustrates the cell wall of ETEC with the inner and outer membrane, LPS, and fimbriae. (**b**) Demonstrates attachment of fimbria on the tip of microvillus and subsequent release of virulence factors. Mucus is degraded by the action of mucinases YghJ and EatA. This action enables ETEC to penetrate the mucosal layer and enabling the tip adhesin to adhere to the microvillus and allow access for LT and ST. (**c**) Flagellum with an EtpA adhesin bound at its tip attaches to the N-acetyl-galactosamine receptor on the microvillus of epithelial cells. In addition, the binding of CFA/I fimbrial tip adhesin CfaE to asialo GM1 on the microvillus is also shown. (**d**) Attachment of ST to guanylyl cyclase receptor and LT to ganglioside GM1 receptor induces secretion of ions into the intestinal lumen, causing diarrhea.

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
