# Peer review of "Human Mucosal IgA Immune Responses against Enterotoxigenic Escherichia coli"

_pathogens, 2020, doi:10.3390/pathogens9090714_

Round 1
Reviewer 1 Report
The present work of Riaz, Steinsland & Hanevik entitled "Human mucosal IgA immune responses against enterotoxigenic Escherichia coli” (ETEC) (ID: pathogens-886082) is a comprehensive literature review of great value to researchers working on the development of vaccines against ETEC and possibly other enteric pathogens.
The interrogation of the mucosal IgA response after ETEC vaccination and/or infection is problematic due to the limitations to obtain samples directly from the gut. Hence, alternative methodologies and approaches have been explored, such as serum ELISA, ELISPOT, measurement of antibodies in ALS and saliva. The manuscript is well and orderly written, and gives the reader a broad view of the topic.
These are my comments and suggestions to authors’ consideration:
- Line 169: The fourth paragraph under “SIgA response to ETEC infection,” is confusing. Please, make sure to consistently refer to “experimental infection” when appropriate. The aforementioned paragraph seems to refer to experimental and natural infection interchangeably, which will be confusing to readers.
- When describing “Direct and indirect measurement of small intestinal SIgA responses,” the authors could emphasize the need for universal standardized protocols (for ELISA, ALS, saliva, fecal etc) to overcome methodological caveats such as the type of plate (for ELISA), concentration of Ag for coating, secondary antibodies, titer calculation, number of cells and incubation time (for ALS) etc. I believe researchers would greatly benefit from using the same protocols. For example, not all laboratories utilize ELISA against the secretory component of the IgA, rather, I believe, the majority employs anti-IgA in their ELISA. In the past, TRF assays seem to have offered different degrees of sensitivity compared to ELISA (https://pubmed.ncbi.nlm.nih.gov/16428745/), and finally, more recently, some groups started using eletrochemiluminescence (MSD), which has advantages over ELISA and can also lead to different response profiles given its higher sensitivity.
- Line 162: As the authors describe, piglets and rabbits are not “readily” colonized by human ETEC… I believe the paragraph can be expanded to describe, exactly, how piglets and rabbits were employed in those studies so readers do not get the wrong impression.
- Line 258 – Please, review the following manuscript about the binding characteristics of CfaB <https://iai.asm.org/content/84/5/1642>. While Ab against CfaB might indeed disrupt binding, the mechanism seems to be interference with the fimbria flexibility rather than directly neutralizing a potential binding site.
- Line 396 – Please, make sure to indicate that cow colostrum contains, mostly, IgG, not IgA. Reference #114 is, indeed, about employing SIgA passive immunization, but the use of cow’s colostrum (IgG) is more commonly referred as oral prophylaxis (which may only last a few hours after administration).
- Unfortunately, a recent oral challenge study with the CS6+ ETEC strain B7A has not yet been published (it is under review), hence it was missed in the literature search; fortunately, more information about it can be found here <https://www.astmh.org/ASTMH/media/2017-Annual-Meeting/ASTMH-2017-Abstract-Book.pdf> (abstract #1759) as well as in the Clinical Trial registry <https://clinicaltrials.gov/ct2/show/results/NCT02773446>. One of the main observations from the study is the lack of protection upon the re-challenge with the B7A strain, which is in opposition to the previous findings of Levine et al (1979).
Author Response
The present work of Riaz, Steinsland & Hanevik entitled "Human mucosal IgA immune responses against enterotoxigenic Escherichia coli” (ETEC) (ID: pathogens-886082) is a comprehensive literature review of great value to researchers working on the development of vaccines against ETEC and possibly other enteric pathogens.
The interrogation of the mucosal IgA response after ETEC vaccination and/or infection is problematic due to the limitations to obtain samples directly from the gut. Hence, alternative methodologies and approaches have been explored, such as serum ELISA, ELISPOT, measurement of antibodies in ALS and saliva. The manuscript is well and orderly written, and gives the reader a broad view of the topic.
Answer: We thank the reviewer for this generally positive assessment
These are my comments and suggestions to authors’ consideration:
Line 169: The fourth paragraph under “SIgA response to ETEC infection,” is confusing. Please, make sure to consistently refer to “experimental infection” when appropriate. The aforementioned paragraph seems to refer to experimental and natural infection interchangeably, which will be confusing to readers.
Answer: We thank the reviewer for pointing this out for us. This paragraph indeed used experimental and natural infections interchangeably, even though, this was not the intention and it has been corrected now. Refer line 186.
When describing “Direct and indirect measurement of small intestinal SIgA responses,” the authors could emphasize the need for universal standardized protocols (for ELISA, ALS, saliva, fecal etc) to overcome methodological caveats such as the type of plate (for ELISA), concentration of Ag for coating, secondary antibodies, titer calculation, number of cells and incubation time (for ALS) etc. I believe researchers would greatly benefit from using the same protocols. For example, not all laboratories utilize ELISA against the secretory component of the IgA, rather, I believe, the majority employs anti-IgA in their ELISA. In the past, TRF assays seem to have offered different degrees of sensitivity compared to ELISA (https://pubmed.ncbi.nlm.nih.gov/16428745/), and finally, more recently, some groups started using eletrochemiluminescence (MSD), which has advantages over ELISA and can also lead to different response profiles given its higher sensitivity.
Answer: We agree with the reviewer that a more standardized way to measure immune responses would benefit the field, and have now included this aspect, and mention the TRF and eletrochemiluminescence methods in accordance with the reviewer’s suggestions. We also take the opportunity to mention how fold increases and responder ratios are used to mitigate this problem.
Line 162: As the authors describe, piglets and rabbits are not “readily” colonized by human ETEC… I believe the paragraph can be expanded to describe, exactly, how piglets and rabbits were employed in those studies so readers do not get the wrong impression.
Answer: We thank the reviewer for this notable comment. The literature on animal studies plays a pivotal role in establishing ETEC as enteric pathogens and comprehension of the immune response. But, the literature in animal studies is very vast and beyond the scope of this review which is limited to human mucosal immune response. Since the earlier statement was somewhat incomplete, it has now been modified to keep it in line with the theme of the review. Kindly refer to line 175.
Line 258 – Please, review the following manuscript about the binding characteristics of CfaB <https://iai.asm.org/content/84/5/1642>. While Ab against CfaB might indeed disrupt binding, the mechanism seems to be interference with the fimbria flexibility rather than directly neutralizing a potential binding site.
Answer: We acknowledge the reviewer’s suggestion. It has been corrected now and a reference has been added to support the statement; refer to line number 276.
Line 396 – Please, make sure to indicate that cow colostrum contains, mostly, IgG, not IgA. Reference #114 is, indeed, about employing SIgA passive immunization, but the use of cow’s colostrum (IgG) is more commonly referred as oral prophylaxis (which may only last a few hours after administration).
Answer: We thank the reviewer for this valuable comment. The sentence is now modified for better understanding and the antibodies employed are clearly mentioned. Refer to line number 419.
Unfortunately, a recent oral challenge study with the CS6+ ETEC strain B7A has not yet been published (it is under review), hence it was missed in the literature search; fortunately, more information about it can be found here <https://www.astmh.org/ASTMH/media/2017-Annual-Meeting/ASTMH-2017-Abstract-Book.pdf> (abstract #1759) as well as in the Clinical Trial registry <https://clinicaltrials.gov/ct2/show/results/NCT02773446>. One of the main observations from the study is the lack of protection upon the re-challenge with the B7A strain, which is in opposition to the previous findings of Levine et al (1979).
Answer: We appreciate that the reviewer has shared this information. We have now mentioned this study in the respective section on line number 368 and referenced the poster in the text. Regrettably, we cannot add more information about this study without the permission of the authors.
Reviewer 2 Report
Dear authors,
The review “Human mucosal IgA immune responses against enterotoxigenic E. coli” submitted by Riaz and colleagues summarizes the current knowledge and research efforts on this topic. It is well suited as a starting point for a fine grasp of different aspects. These are:
- Categorization of the intestinal pathotypes
- Function of the GALT with emphasis on the SIgA response
- Introduction of the different antigens suited for vaccination approaches
- Immune responses induced by natural infections and different vaccination strategies.
The figures and tables included are clearly laid out. For the sake of completeness, please include the following references in the manuscript:
Oral administration of an anti-CfaE secretory IgA antibody protects against Enterotoxigenic Escherichia coli diarrheal disease in a nonhuman primate model. Stoppato M, Gaspar C, Regeimbal J, Nunez RG, Giuntini S, Schiller ZA, Gawron MA, Pondish JR, Martin JC 3rd, Schneider MI, Klempner MS, Cavacini LA, Wang Y. Vaccine. 2020 Feb 28;38(10):2333-2339. doi: 10.1016/j.vaccine.2020.01.064. Epub 2020 Jan 31. PMID: 32008877
U-Omp19 from Brucella abortus increases dmLT immunogenicity and improves protection against Escherichia coli heat-labile toxin (LT) oral challenge. Coria LM, Martinez FL, Bruno LA, Pasquevich KA, Cassataro J. Vaccine. 2020 Jul 6;38(32):5027-5035. doi: 10.1016/j.vaccine.2020.05.039. Epub 2020 Jun 11. PMID: 32536545 Free PMC article.
Nevertheless, the review in its current form has many flaws concerning spelling, grammar, formatting and overall style, which is often circuitous and therefore tiring to read. Enclosed please find a reviewed version including the commentaries. Furthermore, I strongly recommend reviewing by a native speaker.

Author Response
Dear authors, The review “Human mucosal IgA immune responses against enterotoxigenic E. coli” submitted by Riaz and colleagues summarizes the current knowledge and research efforts on this topic. It is well suited as a starting point for a fine grasp of different aspects. These are:
Categorization of the intestinal pathotypes
Function of the GALT with emphasis on the SIgA response
Introduction of the different antigens suited for vaccination approaches
Immune responses induced by natural infections and different vaccination strategies.
The figures and tables included are clearly laid out. For the sake of completeness, please include the following references in the manuscript:
Oral administration of an anti-CfaE secretory IgA antibody protects against Enterotoxigenic Escherichia coli diarrheal disease in a nonhuman primate model. Stoppato M, Gaspar C, Regeimbal J, Nunez RG, Giuntini S, Schiller ZA, Gawron MA, Pondish JR, Martin JC 3rd, Schneider MI, Klempner MS, Cavacini LA, Wang Y. Vaccine. 2020 Feb 28;38(10):2333-2339. doi: 10.1016/j.vaccine.2020.01.064. Epub 2020 Jan 31. PMID: 32008877
U-Omp19 from Brucella abortus increases dmLT immunogenicity and improves protection against Escherichia coli heat-labile toxin (LT) oral challenge. Coria LM, Martinez FL, Bruno LA, Pasquevich KA, Cassataro J. Vaccine. 2020 Jul 6;38(32):5027-5035. doi: 10.1016/j.vaccine.2020.05.039. Epub 2020 Jun 11. PMID: 32536545 Free PMC article.
Answer: We thank the reviewer for the generally positive assessment and recommendations for additional references. The said references have been added in their respective sections.
Nevertheless, the review in its current form has many flaws concerning spelling, grammar, formatting and overall style, which is often circuitous and therefore tiring to read. Enclosed please find a reviewed version including the commentaries. Furthermore, I strongly recommend reviewing by a native speaker.
Answer: We thank the reviewer for many noteworthy suggestions to improve the readability. We have included the proposed changes to improve the manuscript. The language has been read by a native English speaker in our group and some extra language improvements have been made.